# Redirection of Care for Neonates with Hypoxic-Ischemic Encephalopathy Receiving Therapeutic Hypothermia

**DOI:** 10.3390/jcm14020317

**Published:** 2025-01-07

**Authors:** Deborah F. L. Gubler, Adriana Wenger, Vinzenz Boos, Rabia Liamlahi, Cornelia Hagmann, Barbara Brotschi, Beate Grass

**Affiliations:** 1Division of Pediatric Palliative Care, University Children’s Hospital Zurich, CH-8032 Zurich, Switzerland; 2Department of Intensive Care and Neonatology, University Children’s Hospital Zurich, CH-8032 Zurich, Switzerland; 3Newborn Research, Department of Neonatology, University Hospital Zurich, CH-8091 Zurich, Switzerland; 4Child Development Center, University Children’s Hospital of Zurich, University of Zurich, CH-8032 Zurich, Switzerland; 5Children’s Research Center, University Children’s Hospital of Zurich, University of Zurich, CH-8032 Zurich, Switzerland

**Keywords:** hypoxic-ischemic encephalopathy, therapeutic hypothermia, death, neonatal palliative care, withdrawal of life-sustaining therapies, redirection of care

## Abstract

**Background/Objectives**: Hypoxic-ischemic encephalopathy (HIE) in late preterm and term neonates accounts for neonatal mortality and unfavorable neurodevelopmental outcomes in survivors despite therapeutic hypothermia (TH) for neuroprotection. The circumstances of death in neonates with HIE, including involvement of neonatal palliative care (NPC) specialists and neurodevelopmental follow-up at 18–24 months in survivors, warrant further evaluation. **Methods**: A retrospective multicenter cohort study including neonates ≥ 35 weeks gestational age with moderate to severe HIE receiving TH, registered in the Swiss National Asphyxia and Cooling Register between 2011 and 2021. Neurodevelopmental follow-up at 18–24 months in survivors was assessed. The groups of survivors and deaths were compared regarding perinatal demographic and HIE data. Prognostic factors leading to redirection of care (ROC) were depicted. **Results**: A total of 137 neonates were included, with 23 (16.8%) deaths and 114 (83.2%) survivors. All but one death (95.7%) occurred after ROC, with death on a median of 3.5 (2–6) days of life. Severe encephalopathy was indicated by a Sarnat score of 3 on admission, seizures were more frequent, and blood lactate values were higher on postnatal days 1 to 4 in neonates who died. Lactate in worst blood gas analysis (unit-adjusted odds ratio 1.25, 95% CI 1.02–1.54, *p* = 0.0352) was the only variable independently associated with ROC. NPC specialists were involved in one case. Of 114 survivors, 88 (77.2%) had neurodevelopmental assessments, and 21 (23.9%) of those had unfavorable outcomes (moderate to severe disability). **Conclusions**: Death in neonates with moderate to severe HIE receiving TH almost exclusively occurred after ROC. Parents thus had to make critical decisions and accompany their neonate at end-of-life within the first week of life. Involvement of NPC specialists is encouraged in ROC so that there is continuity of care for the families whether the neonate survives or not.

## 1. Introduction

Hypoxic ischemic encephalopathy (HIE) is a leading cause of neonatal death, even in high-income countries [1]. To date, therapeutic hypothermia (TH) is the only evidence-based neuroprotective therapy for HIE to reduce mortality and improve neurodevelopmental outcomes [2,3,4,5,6]. Despite TH, HIE often leads to poor neurodevelopmental outcome prognosis, and thus evaluation for withdrawal of life-sustaining therapy (WLST) is a recurrent topic to discuss with families in the context of HIE [7]. Apart from the redirection of care (ROC) in the face of devastating neurodevelopmental prognosis, some neonates with HIE die due to HIE-related multi-organ failure despite the provision of neonatal intensive care [7].

HIE occurs unanticipated and parents thus face a sudden challenge to the life of their expectedly healthy term-born neonate. TH for HIE is the standard of care for moderate to severe HIE in tertiary neonatal intensive care units (NICUs) [2,3,4,5,6]. Neonatal palliative care (NPC) is emerging as a dedicated subspecialty of pediatric palliative care offered in large tertiary neonatal centers [8,9]. However, the involvement of NPC specialists seems underrepresented in the context of neonatal death due to HIE in Switzerland [10].

The primary objective was to review the circumstances of neonatal deaths of neonates with moderate to severe HIE receiving TH between 2011 and 2021 in two tertiary NICUs in Zurich, Switzerland, with standardized access to NPC. The secondary objective was to compare demographics and clinical prognostic parameters of survivors (including neurodevelopmental outcome at 18–24 months) and deaths of moderate to severe HIE receiving TH, registered in the Swiss National Asphyxia and Cooling Register between 2011 and 2021.

## 2. Materials and Methods

This retrospective multicenter cohort study included neonates ≥ 35 weeks gestational age with HIE registered in the Swiss National Asphyxia and Cooling Register. All of them were admitted to the NICU at the University Hospital Zurich or to the neonatal/pediatric ICU of the University Children’s Hospital Zurich (outborn referral center) between January 2011 and December 2021 for TH due to moderate (Sarnat score 2) or severe (Sarnat score 3) HIE [11]. All neonates were treated according to the Swiss National Asphyxia and Cooling Register Protocol [12]. TH with whole-body cooling was initiated within 6 h of birth, targeting a 33.0 °C to 34.0 °C core temperature, and was continued for 72 h. Exclusion criteria for TH according to the national cooling protocol were applied [12,13].

Inclusion required available data in the Swiss National Asphyxia and Cooling Register. Available results of the 18–24 months neurodevelopmental follow-up (NDFU) assessment in survivors were included. Neonates were excluded if further use of data was refused by parents (consent approached during NICU admission) or if there was evidence of syndromic disease.

Maternal and neonatal demographics, data on pregnancy and delivery, and neonatal characteristics during TH were extracted from the Swiss National Asphyxia and Cooling Register. Resuscitation was defined as any form of ventilation necessary at the age of 10 min. Worst blood gas analysis was defined as the worst blood gas results (arterial, venous, or capillary samples) within 60 min after birth, including umbilical gases (Appendix A). The severity of neonatal encephalopathy was graded on admission and every 24 h during TH. Corresponding Thompson scores [14] were converted to Sarnat scores [11] according to Thompson et al. [14]. All cranial magnetic resonance imaging (cMRI) and cranial ultrasound (cUS) recordings were evaluated by experienced pediatric neuro-radiologists. Data collection was amended manually by electronic medical chart review to collect information on the circumstances of WLST and death.

Prognostic factors of unfavorable neurodevelopmental outcomes [15,16,17] that were included in the discussion to redirect care were determined and depicted in three main categories:Severe clinical neurological abnormalities, reflected in persistent Sarnat scores 2 and 3 [11];Severely abnormal neuro-monitoring with electroencephalogram, defined as significantly reduced or no cerebral activity or therapy-refractory clinical or subclinical seizures or status epilepticus [17,18] (amplitude-integrated electroencephalogram was not included);Severely abnormal neuro-imaging (cMRI or cUS) depicting severe hypoxic-ischemic injury [19,20]

Data on NDFU at 18–24 months were extracted from the Swiss National Follow-up Register. NDFU assessments were performed by an experienced consultant developmental pediatrician and included a clinical examination, a structured neurological assessment that encompassed hearing and vision, and a developmental assessment, utilizing either the Bayley Scales of Infant and Toddler Development, Third Edition (BSID-III) [21] or the Griffiths Development Scales [22], following the guidelines of the two follow-up centers. BSID-III testing provided three main composite scores: cognitive (CCS), language (LCS), and motor composite score (MCS), with a mean score of 100 and a SD of ±15 [21]. The Griffiths Development Scales (Griffiths) comprised five subscales: locomotor, personal-social, hearing and speech, eye and hand coordination, and performance. Results were declared in months (developmental age). In order to compare Griffiths with BSID-III, a developmental quotient was calculated for each domain of the Griffiths (developmental age in relation to adjusted chronological age). Cerebral palsy was graded according to the Gross Motor Function Classification System (GMFCS) of Palisano et al. for children aged ≤2 years [23].

Neurodevelopmental outcomes at 18–24 months were categorized as follows [2,3,4,5,24]:

Unfavorable outcome:Death;Severe disability, defined as a CCS, LCS, or MCS (BSID-III); one Griffiths developmental quotient more than 2 SD below the mean score (i.e., <70); a GMFCS grade of level 3 to 5; hearing impairment (inability to follow commands despite amplification); or blindness;Moderate disability, defined as a CCS, LCS, or MCS (BSID-III) or one Griffiths developmental quotient of 1 to 2 SD below the mean score (i.e., 70 to 84) in addition to one or more of the following: GMFCS grade of level 2; hearing impairment (hearing deficit with the ability to follow commands after amplification); or persistent seizure disorder.

Favorable outcome:Mild disability, defined as a CCS, LCS, or MCS (BSID-III); one Griffiths developmental quotient of 70–84 alone; a CCS or LCS ≥ 85 and GMFCS level 1 or 2; seizure disorder without anti-epileptic medication; or hearing deficit with ability to follow commands with amplification;No disability: absence of previously listed disabilities.

All missing patient information was reported. All statistical analyses were conducted using R v4.4.1 (The R Foundation for Statistical Computing, Vienna, Austria). Descriptive data were reported as frequency (proportion) for categorical variables and as median (interquartile range) for continuous variables. Subgroups for the main outcomes death and survival were compared using Pearson’s Chi-squared and Fisher’s exact test for dichotomous variables and a Wilcoxon rank sum test for continuous variables in unadjusted analysis. Binary logistic regression analysis was used to explore associations between maternal or neonatal characteristics and the main outcomes, and adjusted odds ratios with corresponding 95% confidence intervals (CI) were calculated accordingly. Logistic regression analyses were adjusted for Apgar at 1 min, resuscitation in the delivery room, umbilical artery blood pH, highest lactate in blood gas analysis, Sarnat score on admission (prior to TH), and seizures on the first postnatal day. Goodness-of-fit was assessed using the Hosmer–Lemeshow test. A *p*-value of <0.05 was considered statistically significant.

Data collection, analysis, and publication were approved by the Swiss ethical committee of the Canton of Zurich (KEK-ZH Number 2024-00208). STROBE guidelines were followed for reporting this observational study.

## 3. Results

### 3.1. Participants and Descriptive Data

A total of 154 neonates were initially eligible. However, two neonates with syndromic diseases and 15 neonates whose parents refused participation were excluded, resulting in a final study population of 137 neonates (Figure 1).

### 3.2. Outcome Data and Main Results

A total of 23 out of 137 (16.8%) neonates died in the NICU. Demographic and maternal characteristics did not differ between neonates who died and those who survived (Table 1 and Table 2).

Neonates who died had lower Apgar scores at 1 (0 vs. 2, *p* = 0.0003), 5 (1 vs. 4, *p* = 0.0032), and 10 (2 vs. 5, *p* = 0.0056) minutes after birth as compared to survivors. Blood gas parameters revealed more severe acidosis in those who died (Table 2).

The majority of neonates who died had a Sarnat score of 3 before and during hypothermia, whereas most survivors had a Sarnat score of 2. Seizures were more prevalent in neonates who did not survive on day 1 (47.8% vs. 16.7%, *p* = 0.0022), day 2 (33.3% vs. 12.3%, *p* = 0.0227), day 3 (42.9% vs. 6.1%, *p* = 0.0006), and in total (60.9% vs. 24.6%, *p* = 0.0014). Cerebral MRI scans were performed in almost all survivors, whereas the rate was low in neonates who died (97.4% vs. 30.4%, *p* < 0.0001). Daily highest lactate values were higher during postnatal days 1 to 4 in neonates that were deceased (Table 3a).

Of the 23 neonates who died, one neonate passed away on the first postnatal day despite intensive care interventions due to cardiorespiratory failure. Twenty-two (95.7%) neonates died after ROC. In addition to clinical neurological abnormalities according to the Sarnat score, severely abnormal electroencephalogram (63.4%) [significantly reduced or no cerebral activity], seizures (59.1%) [therapy-refractory clinical or subclinical seizures or status epilepticus], severely abnormal cMRI (31.8%), and pathologic cUS (95.5%) [depicting severe hypoxic-ischemic injury] were reasons to redirect care. The decision to redirect care was made on median postnatal day 3 (2–5), and the neonates eventually died on median day 3.5 (2–6) of life, resulting in a time period of 0 (0–1) days from the decision to redirect care to death (Figure 2 and Table 3b).

The multiprofessional NPC team was involved in one (4.5%) case of ROC. Between the two centers studied, there were no differences in the frequency of ROC (13.3% vs. 17.6%, *p* = 0.6997), the age at the decision to redirect care (2.5 vs. 3.5 days, *p* = 0.5434), and the time period from the decision to redirect care to death (0 vs. 0 days, *p* = 0.7058).

A total of 88 (77.2%) of 114 surviving infants had neurodevelopmental assessments (n = 111 BSID-III, n = 3 Griffiths) at the age of 24 (23–25) months. Among these, 37 (42.0%) infants had no disability, 30 (34.1%) infants had mild disability, 11 (12.5%) infants had moderate disability, and 10 (11.4%) infants had severe disability. Among the entire study cohort, 44 (32.1%) neonates had an unfavorable outcome, and among the survivors, 21 (23.9%) had a moderate or severe disability.

In the adjusted analysis, the decision to redirect care was independently associated with lactate in the worst blood gas analysis (unit adjusted odds ratio 1.25, 95% CI 1.02–1.54, *p* = 0.0352). The logistic regression model fits well to predict the ROC (Hosmer–Lemeshow *p* = 0.920).

## 4. Discussion

This study revealed that almost all deaths in moderate to severe HIE receiving TH in the examined Swiss NICUs occurred in the context of WLST during or shortly after TH. The decision for ROC was made due to severe clinical neurological abnormalities, severely abnormal neuro-monitoring, and neuro-imaging, prognosticating an unfavorable neurodevelopmental outcome [15,16,17,18,19,20]. In 76% of the survivors, neurodevelopmental follow-up assessment at the age of two years confirmed favorable outcomes.

There was no difference in maternal and demographic characteristics between survivors and deaths. As expected, deceased neonates suffered from more severe HIE. This was confirmed by lower Apgar scores and blood gas parameters revealed more severe acidosis (and persistent lactate) compared to survivors, which is in line with previous reports [15,16]. Clinical encephalopathy was more severe (i.e., Sarnat score 3), often accompanied by seizures, in deaths compared to survivors.

Mortality was 17% in this study, which is comparable to an overall median mortality of 14% in moderate to severe HIE receiving TH in Switzerland [13]. Internationally, mortality rates of 14% (Canada) [7] to 27% (France) [17] to 31% (Turkey and Iran) [15,16] were reported. This Swiss cohort of HIE compared best to the Canadian multicenter data registry study (level IV NICUs), evaluating the frequency and timing of WLST in moderate or severe HIE [7]. A total of 96% of deaths occurred after ROC in this study, which is comparable to 88% in the Canadian cohort [7]. Moreover, 94% of parents of a neonate with severe HIE had “treatment limitation” discussions [7,26].

However, within Switzerland, mortality between cooling centers ranged from 0 to 25% in the national study of the Swiss National Asphyxia and Cooling Register on short-term outcomes [13]. Deveci et al. also reported mortality rates of 10–60% in Turkey, with a national average of 23% [16]. These differences in mortality are unlikely to be explained by deviating medical aspects in the care of HIE, especially since all Swiss cooling centers follow the same National Asphyxia and Cooling Protocol. These perceived discrepancies underline the importance of non-medical aspects (ethics, social attitude) in ROC evaluations.

As all but one neonate died after ROC, psychosocial aspects surrounding this decision need to be considered. The centers studied both follow a shared decision-making approach when counseling parents of neonates with HIE with expected poor neurodevelopmental prognosis. The discussion of finding a “threshold” for redirecting care versus accepting less favorable neurodevelopmental outcomes was beyond the scope of this report.

The socioeconomic status score of parents did not differ between the groups. Other socioeconomic factors associated with ROC according to Natarajan et al., such as health insurance coverage and distant geographics, do not apply to Switzerland [7]. In this study, there was no center difference in the rate of ROC and timing was similar, in line with the Canadian report (day of life 3.5 (2–6) versus 4.0 (2–8)) [7].

In about two-thirds of deaths, cMRI was not performed prior to ROC. However, neuro-imaging with cUS and neuro-monitoring with amplitude-integrated electroencephalogram were performed on all neonates prior to ROC. Most neonates also had an electroencephalogram recording before redirecting care. Interestingly, in the Canadian cohort, 57% of neonates had no cMRI and 25% no (amplitude-integrated) electroencephalogram prior to WLST, and these examinations were not considered necessary to initiate end-of-life discussions with parents [7].

In this study, NPC specialists were involved in one case of ROC only. While an approach for many years in some countries has been to initiate palliative care from the point of diagnosis or recognition, in the corresponding healthcare setting the decision-making process in regards to referral to a NPC specialist has been more individual [27,28,29,30]. Team resources can be a restraint in countries in which NPC is still developing and core palliative care is provided by the NICU team [31,32]. In ROC, involving NPC specialists early in the NICU would facilitate continuous care and support NICU staff and families to create memories, have close ones present, and perform rituals or baptisms [33]. NPC teams provide ongoing grief support and optimized bereavement follow-up of families whose neonates died in the NICU, including sibling care [31,33].

All cases of moderate to severe HIE receiving TH should be reviewed systematically in a multidisciplinary, multiprofessional team meeting to reach a consensus for a neurodevelopmental prognosis [16], including prognostic indicators for ROC [34]. In some settings, admission of a neonate with HIE to the NICU is an indication for involving NPC specialists [27,31]. Consultation of NPC and comprehensive parental counseling with the involvement of NPC specialists [35] is less standardized for parents of neonates with HIE in Switzerland [10,36]. There is still an unmet need for continuing NPC support both in ROC in neonates with HIE and in HIE survivors during and after NICU discharge [10]. In view of an increasing prevalence of perinatal life-limiting conditions, guidelines for NPC in the NICU are strongly recommended [37].

This multicenter study benefited from the prospective data entry in the Swiss National Asphyxia and Cooling Register over a decade with the application of the standardized national cooling protocol. The cohort included a high proportion of outborns with HIE, which possibly limits generalizability to settings with inborns. Data on 18–24 months of neurodevelopmental follow-up assessment were retrospectively available for 77% of survivors. However, assessors were not blinded since follow-up occurred as routine clinical care and was limited to 18–24 months. The documentation of end-of-life discussions with families in the medical charts varied notably, and data on counseling were thus not included in the analysis. The findings of this study, especially on ROC, should be interpreted mainly in the context of Swiss NICUs and Swiss society; however, some results seemed generalizable to comparable high-income health settings.

## 5. Conclusions

Death in neonates with moderate to severe HIE receiving TH mainly occurred in the face of expected devastating neurodevelopmental prognosis, leading to WLST in the NICU. Involvement of NPC in cases of moderate and severe HIE, particularly in end-of-life decisions and ROC, warrants national guidelines endorsed by neonatologists and NPC specialists.

## Figures and Tables

**Figure 1 jcm-14-00317-f001:**
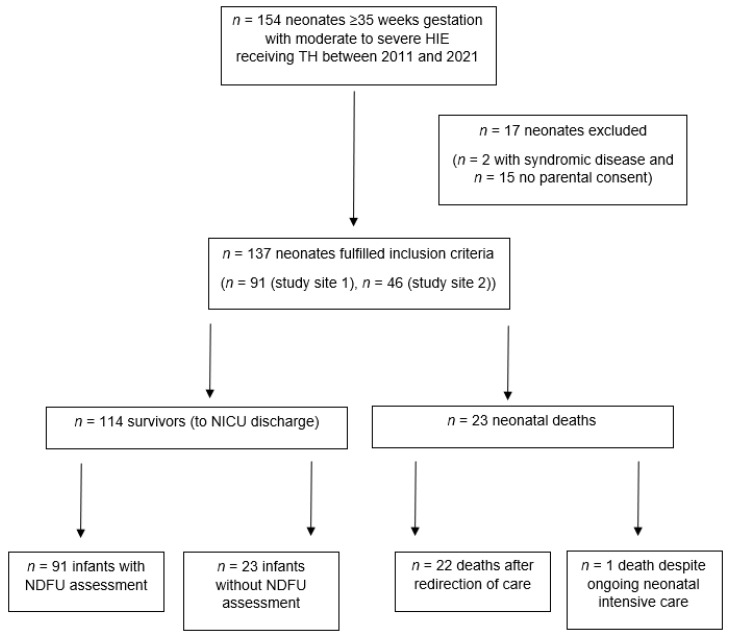
Flowchart of included neonates. HIE: hypoxic-ischemic encephalopathy; TH: therapeutic hypothermia; NICU: Neonatal Intensive Care Unit; NDFU: neurodevelopmental follow-up assessment at 18–24 months of age.

**Figure 2 jcm-14-00317-f002:**
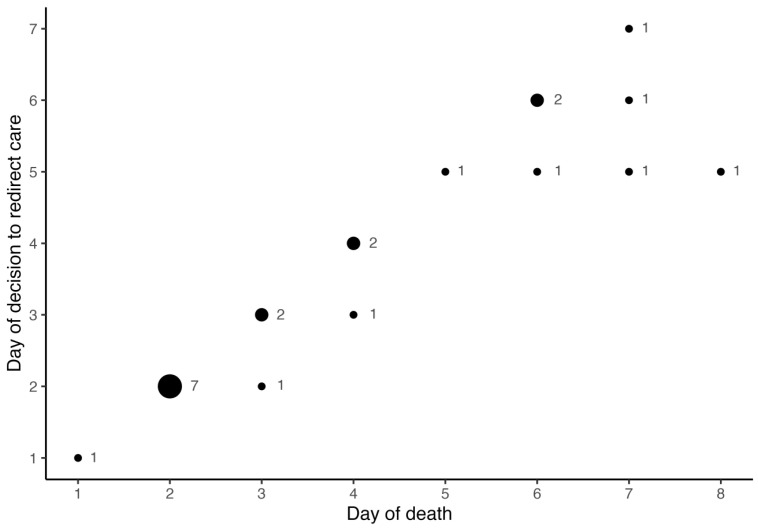
Comparison of age at decision to redirect care and age at death. Age (postnatal days) at decision to redirect care and occurrence of death. The size of the circle refers to the absolute number of patients (added next to each circle).

**Table 1 jcm-14-00317-t001:** Maternal, pregnancy, and delivery characteristics.

Characteristics	Death (n = 23)	Survivor (n = 114)	
	n (%) or Median (IQR)	n (%) or Median (IQR)	*p*-Value
Sex, male	14 (60.9)	58 (50.9)	0.5179
Gestational age (weeks)	40.0 (39.0, 40.6)	39.8 (38.4, 40.7)	0.5038
Birth weight (grams)	3450 (3050, 3658)	3380 (3000, 3724)	0.5542
Outborn	23 (100.0)	105 (92.1)	0.3561
Small for gestation (<10. percentile)	2 (8.7)	14 (12.3)	1.0000
Multiple gestation	1 (4.3)	4 (3.5)	1.0000
Maternal age (years)	34 (31, 38)	33 (30, 35)	0.2930
Primiparae	15 (71.4)	68 (62.4)	0.5879
SES Score	4 (2, 10)	4 (3, 6)	0.8834
Mode of delivery			
Vaginal delivery (all)	12 (52.2)	61 (53.5)	1.0000
spontaneous, cephalic	7 (30.4)	36 (31.6)	
spontaneous, breech	0 (0.0)	2 (1.8)	
Instrumental	5 (21.7)	23 (20.2)	
Caesarean delivery (all)	11 (47.8)	53 (46.5)	1.0000
elective CS	1 (4.3)	5 (4.4)	
emergency CS	8 (34.8)	41 (36.0)	
secondary CS	2 (8.7)	7 (6.1)	

CS: caesarean section; SES: socioeconomic status [25] [Largo]. Missing values: maternal age = 3 and 19, primiparae = 2 and 5, and SES score = 5 and 24, in the death and survivor group, respectively.

**Table 2 jcm-14-00317-t002:** Neonatal characteristics.

Characteristics	Death (n = 23)	Survivor (n = 114)	
(According to Swiss National Cooling Protocol)	n (%) or Median (IQR)	n (%) or Median (IQR)	*p*-Value
Apgar score			
at 1 min	0 (0, 1)	2 (1, 3)	0.0003
at 5 min	1 (1, 4)	4 (2, 5)	0.0032
at 10 min	2 (1, 4)	5 (3, 6)	0.0056
Resuscitation in the delivery room *	18 (78.3)	65 (57.0)	0.0650
Umbilical artery pH	6.87 (6.79, 7.04)	6.99 (6.88, 7.10)	0.0934
Blood gas analysis, worst ^#^			
pH	6.78 (6.61, 6.80)	6.86 (6.79, 6.95)	<0.0001
Base deficit (mmol/L)	24.4 (19.1, 28.5)	19.0 (14.8, 22.0)	0.0029
Lactate (mmol/L)	17.7 (15.0, 19.0)	14.1 (11.5, 16.0)	0.0003

* Resuscitation was defined as any form of ventilation necessary at the age of 10 min. ^#^ Worst blood gas analysis was defined as worst blood gas results within 60 min after birth, including umbilical gases. Missing values: Apgar at 1 min = 1 and 2, Apgar at 5 min = 2 and 1, Apgar at 10 min = 2 and 1, umbilical artery pH = 5 and 15, pH = 1 and 5, base deficit = 6 and 12, lactate = 4 and 17, in the death and survivor group, respectively.

**Table 3 jcm-14-00317-t003:** Prognostic factors of expected unfavorable neurodevelopmental outcome leading to redirection of care. (**a**). Comparison of prognostic factors between deaths and survivors. (**b**). Depiction of prognostic factors in cases of death.

(**a**)													
**Neurologic Variables**							**Death (n = 23)**				**Survivor (n = 114)**		
							**n (%) or Median (IQR)**				**n (%) or Median (IQR)**		***p*-Value**
Sarnat Score													
on admission (prior to TH)													0.0177
stage 1							0 (0.0)				5 (4.5)		
stage 2							9 (39.1)				73 (65.8)		
stage 3							14 (60.9)				33 (29.7)		
day 1													<0.0001
stage 1							1 (5.0)				29 (26.9)		
stage 2							6 (30.0)				62 (57.4)		
stage 3							13 (65.0)				17 (15.7)		
day 2 *													<0.0001
stage 1							1 (4.8)				33 (31.7)		
stage 2							4 (19.0)				61 (58.7)		
stage 3							9 (42.9)				10 (9.6)		
day 3 *													<0.0001
stage 1							1 (7.1)				38 (38.8)		
stage 2							2 (14.3)				53 (54.1)		
stage 3							7 (50.0)				7 (7.1)		
day 4 *													0.0034
stage 1							1 (9.1)				45 (50.0)		
stage 2							4 (36.4)				39 (43.3)		
stage 3							4 (36.4)				6 (6.7)		
Seizure													
day 1							11 (47.8)				19 (16.7)		0.0022
day 2 *							7 (33.3)				14 (12.3)		0.0227
day 3 *							6 (42.9)				7 (6.1)		0.0006
day 4 *							3 (27.3)				7 (6.1)		0.0436
Total							14 (60.9)				28 (24.6)		0.0014
MRI done at discharge							7 (30.4)				111 (97.4)		<0.0001
Highest lactate (mmol/L)													
day 1							11.0 (10.5, 15.2)				6.8 (4.3, 12.0)		0.0233
day 2 *							7.3 (5.5, 7.8)				3.4 (2.2, 4.7)		0.0063
day 3 *							4.3 (3.7, 4.6)				2.0 (1.5, 3.4)		0.0245
day 4 *							3.0 (2.6, 4.4)				1.5 (1.2, 2.1)		0.0297
(**b**)													
**Case**	**Death**		**EEG**	**Seizure**	**cMRI**	**cUS**	**Sarnat Score**				**ROC**		**Death (DOL)**
	**Despite Maximal Support**	**After ROC**					**on Admission**	**DOL 1**	**DOL 2**	**DOL 3**	**Decision (DOL)**	**Execution (DOL)**	
1	0	1	0	0	0	1	3	3	NA	NA	1	1	1
2	0	1	1	0	1	1	3	3	3	3	6	6	6
3	0	1	1	1	0	1	2	3	2	2	5	5	5
4	0	1	0	1	0	0	3	3		NA	2	2	2
5	0	1	0	1	0	1	2			NA	2	2	2
6	0	1	0	1	0	1	3		3	NA	2	2	2
7	0	1	0	0	0	1	3	3		NA	2	2	2
8	0	1	1	1	1	1	2	2	2	3	5	6	6
9	0	1	1	1	1	1	3	3	3	3	5	7	8
10	0	1	1	1	1	1	2	2	3	3	6	7	7
11	0	1	0	1	0	1	2	2	3	3	4	4	4
12	0	1	0	1	0	1	3	3	3		3	3	3
13	0	1	1	1	0	1	3			NA	2	2	2
14	0	1	1	0	0	1	3	3		NA	2	2	2
15	0	1	1	0	0	1	3	3		NA	2	2	3
16	0	1	1	1	1	1	3	2	2	2	7	7	7
17	0	1	1	0	0	1	2	1	1	1	6	6	6
18	0	1	1	1	0	1	2	2	2		3	3	3
19	0	1	0	0	0	1	3	3		NA	2	2	2
20	0	1	1	0	0	1	2	3	3		3	4	4
21	0	1	1	0	1	1	3	3	3	3	4	4	4
22	0	1	1	1	1	1	2	2	3	3	5	7	7
23	1	0	0	0	0	0	3	3	NA	NA	NA	NA	1

(**a**) TH: therapeutic hypothermia * The number of deaths on the first, second, and third day of life was 2, 7, and 3, respectively. Therefore, the death group on the second, third, and fourth day of life contains only 21, 14, and 11 surviving infants, respectively. Missing values: Sarnat before cooling = 3 in the survivor group. Sarnat day 1 = 3 and 6, Sarnat day 2 = 7 and 10, Sarnat day 3 = 4 and 16, Sarnat day 4 = 2 and 24, highest lactate on day 1 = 16 and 48, day 2 = 15 and 48, day 3 = 10 and 49, and day 4 = 8 and 51 in the death and survivor group, respectively. (**b**) DOL: day of life; ROC: redirection of care; NA: not applicable. Empty fields correspond to missing values. EEG: severely abnormal electroencephalogram (EEG), defined as significantly reduced or no cerebral activity. Seizure: therapy-refractory clinical or subclinical seizures or status epilepticus. cMRI: cranial magnetic resonance imaging depicting severe hypoxic-ischemic injury. cUS: cranial ultrasound depicting severe hypoxic-ischemic injury (note: continuous amplitude-integrated EEG monitoring in all cases, not reported).

## Data Availability

The datasets used and/or analyzed during the current study are available from the corresponding author upon reasonable request.

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
