# Peer review of "Redirection of Care for Neonates with Hypoxic-Ischemic Encephalopathy Receiving Therapeutic Hypothermia"

_jcm, 2025, doi:10.3390/jcm14020317_

Round 1

Reviewer 1 Report

Comments and Suggestions for Authors

My comments/suggestions are as described below:

1. Abstract:

a. Line no 18: Change "(near) term" to "late preterm and term"

b. While the methods and results have information about the 2-year follow-up data, the objectives section does not allude to the long-term follow-up. This should be rectified.

c. The last sentence of the abstract should be restructured (e.g. Involvement of NPC specialists is encouraged in ROC so that there is continuity of care for the families whether the child survives or not.)

2. Introduction:

a. Line no 47: Why the name Natarajan is included with ref no 7?

b. Objectives of the study to also include reporting on the long-term follow-up data

3. Results: 

a. In section 3.1, it says that 15 neonates were excluded as the parents refused participation. I would like to know how were the parents approached as this was a retrospective study spanning over 11 years.

b. Table 1, last row: what is meant by secondary CS?

c. Table 3: The legend says that the death group on 2nd, 3rd and 4th day of life contains only 21, 14 and 11 surviving infants. However, in the table, on day 1, there are 20 infants with Sarnat staging in the death column and on day 3, there are 10 infants with Sarnat staging in the death column.

d. For figure 2, it is difficult to comprehend the actual number from the size of the circle. It would be better if the circle is replaced/accompanied by the actual number.

4. Discussion:

a. 4th paragraph: insert "to be" between "unlikely" and "explained". The last sentence of this paragraph should be explored further (what non-medical aspects are the authors referring to?)

b. 9th paragraph: no brackets needed for the word "neurodevelopmental"

c. Discussion also does not talk about the 2-year data. I am wondering if the collection of the 2-year data was an afterthought for this study.

d. Discussion should include a comment that the majority of the infants were outborn, why that was so and how that might impact the results and the study generalizability.

Author Response

Please see attachment, thank you.

Reviewer 2 Report

Comments and Suggestions for Authors

General Comments

This is a well written manuscript addressing an important area.  The authors examined the mortality rates resulting from redirection of care in newborn infants after exposure to hypoxic-ischemic encephalopathy, who were being treated with therapeutic hypothermia. The authors concluded that mortality resulting from redirection of care in newborns exposed to moderate to severe hypoxic-ischemic encephalopathy, who were being treated with therapeutic hypothermia, occurred predominantly, when the neurodevelopmental outcomes were anticipated to result in a devastating sequala. 

However, there are several limitations to the study. The study is mostly relevant to treatment in Switzerland, perhaps reflecting other high-income countries. Inclusion of the outcomes in a wider distribution of NICUs outside Switzerland would increase the generalizability of the study. 

In order for the follow-up data to be reliable, the follow-up rate should exceed 90 percent and examiners should have specific training in the follow-up testing to ensure reliability among the different examiners and sites to ensure the accuracy of the follow-up data. Moreover, the study encompassed a 10-year interval. Were there any changes over time?

Specific Comments

1. There are far too many abbreviations for a short manuscript.

Author Response

Please see attachment, thank you.
